# The Role of Mindfulness in Mitigating the Detrimental Effects of Harsh Parenting among Chinese Adolescents: Testing a Moderated Mediation Model in a Three-Wave Study

**DOI:** 10.3390/ijerph19159731

**Published:** 2022-08-07

**Authors:** Wenyan Sun, Tengfei Guo, Karen Spruyt, Zhijun Liu

**Affiliations:** 1School of Management, Zunyi Medical University, Zunyi 563000, China; 2School of Vocational Education, Guangdong Polytechnic Normal University, Guangzhou 510665, China; 3NeuroDiderot, INSERM, Université de Paris, 75019 Paris, France

**Keywords:** harsh parenting, depressive symptoms, suicidal ideation, mindfulness, Chinese adolescents

## Abstract

Based on the conservation of resources theory, this study aimed to investigate the mediating role of depressive symptoms and the moderating role of mindfulness in the association between harsh parenting and adolescent suicidal ideation in the Chinese cultural context. Using a three-wave (i.e., three months apart) data collection among 371 Chinese adolescents, this study found that depressive symptoms mediated the relationship between harsh parenting and adolescent suicidal ideation. Moreover, adolescent mindfulness mitigated the effects of harsh parenting on suicidal ideation, as well as the indirect effect of harsh parenting on suicidal ideation via depressive symptoms. The theoretical and practical implications are discussed.

## 1. Introduction

Adolescence is the transition period from childhood to adulthood, in which adolescents may experience various mental health problems due to their vulnerability and lability [1]. The literature indicates that suicidal behavior [2], depressive symptoms [3], and aggression are common mental health problems in adolescence [4], of which suicidal behavior has been widely discussed by scholars and practitioners because of its high case-fatality ratio [5]. Although numerous studies on adolescent suicide have been conducted, its causes and mechanisms remain unclear in different cultural backgrounds [6,7]. In order to effectively prevent and control adolescent suicide, it is necessary to investigate this topic in China, which is an Eastern cultural context with a large population [8].

Suicidal ideation is an invasive and repetitive way of thinking regarding death or harming oneself, which is highly likely to lead adolescents to transform abstract ideas or thoughts into specific plans [9,10,11]. In the suicidal process of adolescents, suicidal ideation has been considered one of the major factors causing the suicide of adolescents [8,12]. Through a survey of 24,345 Chinese adolescents, Guo et al. found that 13.7% of adolescents experience suicidal ideation [13]. Therefore, society must pay more attention to the Chinese adolescent suicidal ideation problem [8,13]. Existing studies have revealed that risk factors such as an unsupportive family environment [2], drug abuse [14], and bullying may affect the suicidal ideation of adolescents, of which an unsupportive family environment plays a crucial role in its formation [15,16]. Low [17] revealed that a negative parenting style often leads to a higher risk of suicide in adolescents. Unfortunately, how harsh parenting affects the suicidal ideation of Chinese adolescents has not been fully investigated, which necessitates further exploration.

Harsh parenting is a typical form of negative parenting style, defined as physical and verbal aggression against children, including coercion and discipline of a punitive nature, as well as negative emotional outbursts such as “yelling, verbal abuse, and aggression” [18,19]. The impact of harsh parenting on adolescent mental health has been well documented in existing research. For example, adolescents who experience harsh parenting are more likely to report suicidal ideation [20]. However, most of the relevant research on harsh parenting has focused on families in the United States or other Western countries. A meta-analysis by Gershoff and Grogan-Kaylor indicated that 69.4% of harsh parenting studies were from the United States [21], with the remaining small number of studies from Western Europe, and that there is a paucity of relevant studies based on Eastern cultural backgrounds. One example is that of Ahemaitijiang et al. [22] who validated both positive and negative dimensions of parenting to measure parenting styles in the Chinese cultural context. Therefore, the generalizability of the link between harsh parenting and suicidal ideation in Eastern cultural contexts needs to be further tested.

Chinese culture is heavily influenced by Confucianism, which emphasizes traditional “father–son duties” and a hierarchy between Chinese parents and children [23]. In this context, parents are more likely to impose strict discipline on their children. Thus, harsh parenting may be a more common phenomenon in China compared to Western countries. Considering the negative effects of certain parenting styles on adolescent development [24], Chinese adolescents experiencing harsh parenting may tend to develop suicidal ideation [19]. Therefore, exploring the potential effects of harsh parenting on adolescents may help to gain a deeper and more comprehensive understanding of the causes of suicidal ideation in Chinese adolescents.

The conservation of resources (COR) theory suggests that people always strive to protect and obtain resources that they value [25], and when people perceive that they may lose resources or fail to obtain the expected return after investing resources, adverse emotional reactions will occur [26]. As a source of family stress faced by adolescents, harsh parenting may pose a threat of resources to adolescents (e.g., abuse behaviors) or lead to a lack of resources for adolescents (e.g., emotional neglect and lack of support) [27,28]. According to COR theory, when adolescents experience harsh parenting, they may experience less parental support and generate negative emotions such as depressive symptoms [29]. If the resource loss cannot be effectively blocked and there is no opportunity for them to be compensated for it, the resource loss will proceed with acceleration and form a loss spiral, leading to the enhancement of suicidal ideation among these adolescents [30]. Therefore, the loss spiral principle of the COR theory may provide a useful theoretical framework for understanding the relationship between harsh parenting and suicidal ideation in adolescents.

In response to previous calls for longitudinal studies to further explore the relationship between parenting styles and suicidal ideation [13], this study proposes that harsh parenting, a family stressor, may lead to adolescent depressive symptoms and then trigger suicidal ideation based on COR theory. Moreover, mindfulness, a key personal resource [31,32], may play an important role in mitigating and preventing the loss spiral and thus moderate the effects of harsh parenting on depressive symptoms and suicidal ideation. Therefore, this study theoretically proposes an integrated model to understand the relationship between harsh parenting and suicidal ideation in Chinese adolescents (see Figure 1). Our study further empirically examines the proposed model through a three-wave longitudinal design (i.e., three months apart) with a sample of 371 Chinese adolescents.

### 1.1. The Mediation Role of Depressive Symptoms in the Relationship between Harsh Parenting and Suicidal Ideation

Depressive symptoms are negative emotions caused by rumination and hopelessness due to the inability to cope with external pressure [29]. Previous studies provide strong evidence for the relationship between harsh parenting and depressive symptoms [33]. For example, a meta-analytic study conducted by Pinquart found that corporal punishment is positively related to adolescents’ depressive symptoms [34], and their depressive symptom levels increased over time. In addition, cross-sectional and longitudinal studies consistently show that harsh parenting may reduce adolescents’ expectations of future development, and thus increase their depressive symptoms [19,33].

COR theory provides a useful theoretical framework for understanding the relationship between harsh parenting and depressive symptoms. Adolescents who experienced harsh parenting must spend a lot of time and energy dealing with the negative results of harsh parenting (e.g., proving whether they deserve to be loved, whether their parents are satisfied with them, and whether they have done something wrong), which means that more emotional resources needed to be consumed by adolescents in the process of growing up [35,36]. Adolescents living with harsh parenting over time experience a lack of resources, resulting in adverse emotional reactions such as depressive symptoms [37]. In addition, harsh parenting is associated with limited parental support, making it difficult for these adolescents to obtain additional resources from the family [38], which further aggravates the lack of resources and leads to depressive symptoms [39]. In fact, research has shown that adolescents who experience harsh parenting are more likely to internalize this parenting style into failed support [40] and believe it will be difficult for them to obtain help from their family to deal with external pressure [41]. They form a negative view of themselves, resulting in a sense of low self-worth [17,33]. In view of this, as a stressful situation, harsh parenting is likely to trigger adolescent depressive symptoms.

Increasingly, studies are showing that depression is an important risk factor for suicidal ideation [42], though the relationship is complicated. Some studies argue that suicidal ideation is a depressive symptom; however, Kessler et al. clearly pointed out that they are two distinct constructs [43]. This is because depressive symptoms mainly reflect emotional problems, while suicidal ideation reflects a way of thinking and the formation of future plans. The former is likely a prerequisite for suicidal ideation, which exists before the latter [44]. According to the escape theory of suicide [45,46], adolescents who experience depressive symptoms may not be able to endure the intense sadness and a sense of worthlessness but may have the motivation to end their pain through suicidal ideation [47]. In a study on the relationship between depressive symptoms and suicidal ideation, scholars have suggested that adolescents induce a high level of suicidal ideation and implement it to escape pain because they cannot bear the low sense of self-worth brought about by their depressive symptoms [17]. To sum up, adolescents who experience harsh parenting are more likely to generate depressive symptoms, which may further lead to suicidal ideation. Taken together, this study proposes the following hypothesis:

**Hypothesis** **1.**
*Depressive symptoms will mediate the relationship between harsh parenting and suicidal ideation.*


### 1.2. The Moderating Role of Mindfulness

According to COR theory [25,26], adolescents who have experienced harsh parenting are more likely to experience a resource loss spiral and then generate depressive symptoms, which can then cause suicidal ideation. Recently, research has indicated that individuals can prevent the loss of resources and acquire additional resources through personal level resource positive coping styles, to alleviate the negative impact of external pressure on themselves [48]. Research has suggested that mindfulness may reduce the negative effects of stress and trauma associated with adverse childhood exposures, improve short- and long-term outcomes, and potentially alleviate adverse health outcomes in adulthood [49].

In the present study, we propose that mindfulness, an important personal resource, may play a role in preventing a resource loss spiral and supplementing new resources for adolescents by helping them to break away from old experiences, decrease the automaticity of emotion, and increase their tolerance to negative emotions [50,51]. Accordingly, under the framework of the COR theory, this study introduces the construct of mindfulness as an important boundary condition to alleviate the negative effect of harsh parenting.

#### 1.2.1. The Moderating Effect of Mindfulness on the Relationship between Harsh Parenting and Depressive Symptoms

Among the protective factors to reduce mental health problems, the human ability to meditate may be an important factor. This cognitive process may represent mindfulness, which is stimulated by intentional attention to what is happening and a non-critical attitude [52]. In other words, mindfulness emphasizes openness and acceptance, and people experience all their thoughts and feelings in this process from a position of awareness and a lack of judgment [53,54].

This study assumes that mindfulness may weaken the effect of harsh parenting on depressive symptoms for several reasons. Mindfulness can effectively identify individual automatic thinking and unhealthy behavioral patterns, and assist individuals to form better affective regulation skills, thus enhancing their positive emotions and reducing negative emotions [50,55]. Therefore, mindfulness is likely to alleviate the facilitative effect of harsh parenting on depressive symptoms by encouraging adolescents to objectively experience and accept harsh parenting and regulate their negative emotional responses to harsh parenting [56].

Moreover, adolescents with high levels of mindfulness are inclined to consciously focus on their current experience [57,58], thus decreasing negative thoughts brought about by harsh parenting (e.g., a low sense of value and self-doubt). Mindfulness also helps adolescents to observe internal and external stimuli without judgment or evaluation [59], thereby enhancing their ability to discover additional resources (e.g., other individual resources and external resources) and alleviating the resource loss spiral. More available resources for adolescents are likely to lessen the accumulation of negative emotions and the generation of depressive symptoms [60]. As such, the effect of harsh parenting on depressive symptoms may be weaker in adolescents with high levels of mindfulness. Therefore, this study proposes the following assumption:

**Hypothesis** **2.**
*Mindfulness will moderate the relationship between harsh parenting and depressive symptoms, such that the relationship will be weaker at higher levels of mindfulness.*


#### 1.2.2. The Moderating Effect of Mindfulness on the Relationship between Harsh Parenting and Suicidal Ideation

The present study presumes that adolescent mindfulness can mitigate the effects of harsh parenting on suicidal ideation in two ways. First, mindfulness can help adolescents to observe their thoughts and accept psychological and environmental events without judgment [61], so that they can recognize that the self-doubt and low sense of self-worth brought about by harsh parenting is only a feeling or thought, and not necessarily an accurate representation of the facts, thereby decreasing the adverse results of harsh parenting. This objective and neutral perspective can help adolescents reduce painful experiences generated by the resource loss spiral [62,63], and thus reduce the possibility of suicidal ideation.

Second, mindfulness can help adolescents detach themselves from negative experiences [15], and this detachment can reduce rumination and help relieve distressing experiences caused by harsh parenting, which can lower suicidal ideation. Therefore, mindfulness is expected to weaken the positive relationship between harsh parenting and suicidal ideation.

**Hypothesis** **3.**
*Mindfulness will moderate the relationship between harsh parenting and suicidal ideation, such that the relationship will be weaker at higher levels of mindfulness.*


#### 1.2.3. The Moderating Mediation Effect of Mindfulness

As mentioned above, depressive symptoms may mediate the relationship between harsh parenting and suicidal ideation (Hypothesis 1), and mindfulness is likely to moderate the effects of harsh parenting on depressive symptoms (Hypothesis 2) and suicidal ideation (Hypothesis 3). In combination with these hypotheses, we further propose an integrated first-stage moderated mediated model, in which the indirect effect of harsh parenting on suicidal ideation via depressive symptoms would vary depending on adolescents’ levels of mindfulness. Mindfulness may buffer the indirect effect of harsh parenting on suicidal ideation by reducing automaticity and regulating negative emotions generated by the family stressor. As such, the effect of harsh parenting on suicidal ideation through depressive symptoms would be weaker when adolescent mindfulness is higher.

**Hypothesis** **4.**
*Mindfulness will moderate the indirect effect of the relationship between harsh parenting on suicidal ideation via depressive symptoms, such that the indirect effect will be weaker at higher levels of mindfulness.*


## 2. Methods

### 2.1. Participants and Procedures

The present study recruited 470 adolescents from two junior high schools in central China, using a convenience sampling method. Data were collected via self-administered questionnaires. To reduce the concerns about common method bias and avoid spurious causality [64,65], this study conducted three waves of data collection at three-month intervals. At Time 1, which was October 2021, the participants were asked to complete the measures on harsh parenting and mindfulness. At Time 2 (three months after Time 1), the participants completed the measure of depressive symptoms. At Time 3 (three months after Time 2), the participants completed the measure of suicidal ideation.

To enhance the quality of data collection, the researchers explained the research purpose and schedule to the participants and informed them that their participation was voluntary. Moreover, the participants were assured that all questionnaires would be kept confidential, and all data would be used for scientific research purposes only. This study’s procedure complied with the Declaration of Helsinki. The researchers received permission to conduct the research from the principals of the target schools. We also obtained written informed consent from each participant’s parents.

At Time 1, the researchers distributed 470 questionnaires and of them, 434 questionnaires were usable (92.34% response rate). At Time 2, we distributed questionnaires to those who completed the Time 1 survey, and 426 usable questionnaires were returned (98.15% response rate). At Time 3, we distributed questionnaires to those who had completed both Time 1 and Time 2 surveys, and 412 usable questionnaires were received (96.71% response rate). After matching the questionnaires obtained from the three waves, a final sample of 371 usable questionnaires were included in the analysis. Thus, our overall response rate was 78.94%. The final sample was aged between 12 and 16 (M = 13.98, SD = 0.88), and 183 were boys (49.3%).

### 2.2. Measures

#### 2.2.1. Harsh Parenting at Time 1

Harsh parenting was measured using eight items developed by Simons et al. and validated by Wang [18,66]. Four items were used to assess harsh paternal parenting and four items were used to assess harsh maternal parenting. The participants were asked to indicate how often their parents had behaved as described on a 5-point Likert scale, ranging from 1 (almost never did that) to 5 (almost always did that). A sample item is “When I did something wrong or made my parents angry, he (or she) would use an object to beat me.” Following previous research on Chinese adolescents [67], all items were averaged to generate a single indicator of harsh parenting, with higher scores representing higher levels of harsh parenting. Cronbach’s alpha for the scale in this study was 0.88.

#### 2.2.2. Mindfulness at Time 1

We assessed mindfulness with five items from Brown and Ryan’s Mindful Attention and Awareness Scale, adapted by Hülsheger et al. [32,68]. Ni et al. demonstrated that this scale has excellent reliability and validity for Chinese individuals [51]. A sample item is “I find it difficult to stay focused on what’s happening in the present” (reverse scored). The participants were required to rate on a 6-point Likert scale ranging from 1 (never) to 6 (always). All items were averaged, with higher scores indicating higher mindfulness. Cronbach’s alpha for the scale in this study was 0.91.

#### 2.2.3. Depressive Symptoms at Time 2

Depressive symptoms were measured by using the nine-item Patient Health Questionnaire (PHQ-9) developed by Kroenke et al. [69], which is a widely used depressive symptoms severity instrument. Research on the general Chinese population has shown that the Chinese version of the PHQ-9 has excellent reliability and construct validity [70]. The participants indicated how much a symptom has bothered them over the last 2 weeks on a 4-point Likert scale ranging from 1 (rarely or none of the time) to 4 (nearly every day). A sample item is “Feeling down, depressed, or hopeless.” Higher scores indicate higher depressive symptoms. Cronbach’s alpha for the scale in this study was 0.88.

#### 2.2.4. Suicidal Ideation at Time 3

We assessed suicidal ideation by using the Beck Scale for Suicide Ideation (BSI) [71]. Li et al. demonstrated that this scale has good reliability and validity for Chinese individuals [72]. The participants rated the first five items, which are utilized to filter individuals who have suicidal ideation. Each item was rated on a 3-point scale, with higher scores representing higher suicidal ideation. A sample item is “Wish to live.” Cronbach’s alpha for the scale in this study was 0.90.

### 2.3. Analysis of Data

To examine the mediation effect of depressive symptoms (Hypothesis 1), we utilized Model 4 of the PROCESS macro in SPSS 24.0 [73]. We conducted bootstrapping with 5000 resamples to determine the mediation effect. As suggested by other researchers [74,75], this technique does not assume the normality of the sampling distribution and provides more reliable estimations of the mediation effects. If the bias-corrected bootstrap 95% confidence interval (CI) does not include zero, this indicates a significant mediation effect at the level of α = 0.05.

To test the moderating effects of mindfulness on the relationship between harsh parenting and depressive symptoms (Hypothesis 2), and the relationship between harsh parenting and suicidal ideation (Hypothesis 3), we conducted a hierarchical regression analysis. The predictor (i.e., harsh parenting) and moderator (i.e., mindfulness) were mean-centered to decrease multicollinearity effects. Furthermore, the moderated mediation effect of mindfulness (Hypothesis 4) was tested using Model 8 of the PROCESS macro [73]. We performed bootstrapping with 5000 resamples to verify the significance of the moderated mediation effect.

## 3. Results

### 3.1. Common Method Bias

Since data were collected through self-reports, we performed Harman’s single-factor test to assess the possible common method bias using SPSS 24.0 [64]. The results of the exploratory factor analysis with principal component analysis revealed that no single factor accounted for more than 50% of the variance of all measurement items (the first factor accounted for 30.80% of the variance). Thus, common method bias was not a major concern in the present study.

### 3.2. Descriptive Statistics, Reliability, and Validity

Before hypothesis testing, we conducted confirmatory factor analysis (CFA) with AMOS 20.0 to test the measurement model [76]. The results of the CFA showed that the four-factor model (i.e., harsh parenting, mindfulness, depressive symptoms, and suicidal ideation) fits the data well: χ^2^ = 886.10, *df* = 315, χ^2^/*df* = 2.81, *p* < 0.001; comparative fit index (CFI) = 0.92, incremental fit index (IFI) = 0.92, Tucker–Lewis index (TLI) = 0.91, root mean square error of approximation (RMSEA) = 0.06 (see Table 1). Additional analyses showed that the four-factor model was significantly better than the alternative models, including a three (χ^2^ = 1875.10, *df* = 318, χ^2^/*df* = 5.90, *p* < 0.001; CFI = 0.78, IFI = 0.78, TLI = 0.75, RMSEA = 0.10), a two (χ^2^ = 2924.23.10, *df* = 320, χ^2^/*df* = 9.14, *p* < 0.001; CFI = 0.62, IFI = 0.63, TLI = 0.59, RMSEA = 0.13), and a one-factor model (χ^2^ = 3825.27, *df* = 321, χ^2^/*df* = 11.92, *p* < 0.001; CFI = 0.49, IFI = 0.50, TLI = 0.45, RMSEA = 0.16). Our results support the distinctiveness of the measures in this study.

Table 1 presents the means, standard deviations, and intercorrelations between the study variables. Harsh parenting was positively related to depressive symptoms and was positively associated with suicidal ideation. Depressive symptoms were positively related to suicidal ideation.

### 3.3. Test of Mediating Effects

Hypothesis 1 posited that depressive symptoms would mediate the relationship between harsh parenting and suicidal ideation. As shown in Table 2, harsh parenting positively influenced depressive symptoms (*β =* 0.29, *p* < 0.001; Model 1-1), and positively influenced suicidal ideation (*β =* 0.13, *p* < 0.05; Model 2-2). Depressive symptoms positively predicted suicidal ideation (*β =* 0.30, *p* < 0.001; Model 2-2). Furthermore, the mediation analysis indicated that the indirect effect of harsh parenting on suicidal ideation was statistically significant, as the bias-corrected bootstrap 95% confidence interval excluded zero (indirect effect = 0.06; Boot SE = 0.02; 95% CI = [0.029, 0.110]). Therefore, Hypothesis 1 was supported.

### 3.4. Moderating Effects of Mindfulness

In Hypothesis 2, we proposed that mindfulness would moderate the relationship between harsh parenting and depressive symptoms. As presented in Table 2, the effect of the interaction term (harsh parenting × mindfulness) on depressive symptoms was nonsignificant (*β =*
*0*.01, *p* > 0.05; Model 1-2), which failed to support Hypothesis 2.

Hypothesis 3 posited that mindfulness would moderate the relationship between harsh parenting and suicidal ideation. The results showed that the interaction term (harsh parenting × mindfulness) was significantly associated with suicidal ideation (*β =* −0.13, *p* < 0.05; Model 2-3), supporting the moderating effect of mindfulness. We conducted a simple slope test to determine the direction of the interaction [77]. As shown in Figure 2, the relationship between harsh parenting and suicidal ideation was stronger when mindfulness was low (−1 *SD*) than when it was high (+1 *SD*). For lower mindfulness, the relationship between harsh parenting and suicidal ideation was stronger and positive (*B* = 0.14, *SE* = 0.06, *t* = 2.53, *p* < 0.05), and for higher mindfulness, the relationship between harsh parenting and suicidal ideation was weaker and negative (*B* = −0.09, *SE* = 0.07, *t* = −1.21, *p* > 0.05). Thus, Hypothesis 3 was supported.

### 3.5. Moderated Mediation Effects of Mindfulness

We further proposed that mindfulness would moderate the indirect effect of harsh parenting on suicidal ideation through depressive symptoms in Hypothesis 4. As shown in Table 3, the conditional indirect effect of harsh parenting on suicidal ideation via depressive symptoms was stronger and significant when mindfulness was low (−1 *SD*) (indirect effect = 0.04; 95% CI = [0.001, 0.066]), but insignificant when mindfulness was high (+1 *SD*) (indirect effect = 0.03; 95% CI = [−0.001, 0.065]). Thus, Hypothesis 4 was supported.

## 4. Discussion

Based on the COR theory, we utilized a three-wave study design to empirically examine the mediating role of depressive symptoms and the moderating role of mindfulness in the relationship between harsh parenting and suicidal ideation among Chinese adolescents. We initially revealed how harsh parenting, a family environmental factor, influences adolescents’ suicidal ideation, as well as when harsh parenting exerts a stronger or weaker effect on adolescents’ suicidal ideation. The main findings of this study were as follows: (1) depressive symptoms mediated the relationship between harsh parenting and suicidal ideation, (2) mindfulness buffered the relationship between harsh parenting and suicidal ideation, but it did not moderate the relationship between harsh parenting and depressive symptoms, and (3) mindfulness mitigated the indirect effect of harsh parenting on suicidal ideation through depressive symptoms.

To the best of our knowledge, this is one of the first studies exploring the complex role depressive symptoms and mindfulness play in the relationship between harsh parenting and adolescent suicidal ideation through an integrative model. Our findings extend the existing literature by investigating the interaction between an environmental factor (i.e., harsh parenting) and an individual level factor (i.e., mindfulness) on depressive symptoms and subsequent suicidal ideation among Chinese adolescents, which has significant theoretical and practical implications for the prevention and treatment of suicidal ideation in adolescents.

### 4.1. Theoretical Implications

#### 4.1.1. The Mediation Effect of Depressive Symptoms

Existing research has indicated that suicidal ideation in adolescents is one of the most severe public health problems [78]. This study revealed that depressive symptoms mediate the effect of harsh parenting on suicidal ideation in Chinese adolescents. This helps us to understand why and how harsh parenting triggers suicidal ideation and responds to the call for more empirical research to uncover the underlying mechanism through which harsh parenting leads to adolescents’ mental health problems in a Chinese cultural context [13]. Accordingly, the present study adds to the theoretical and empirical research regarding harsh parenting and suicidal ideation in the Chinese cultural context.

Specifically, we found that harsh parenting led to more depressive symptoms, which is in line with previous findings [19]. Bauer et al. found that harsh parenting and its negative consequences are more common and severe in middle-income countries [1]. Our study supported the negative consequences of harsh parenting in Chinese adolescents. Due to Confucianism and the traditional father–son relationship in China, the hierarchy between parents and children is emphasized, which may increase the possibility of harsh parenting [79,80]. Therefore, harsh parenting is likely in this culture, leading to the development of depressive symptoms. Our study helps us to better understand the formation of depressive symptoms in Chinese adolescents and extends the literature on harsh parenting in an Eastern cultural context [19]. According to the COR theory, adolescents suffering from harsh parenting are likely to deplete more resources to cope with physical and verbal aggression [25,26], and they have difficulties replenishing resources from the family. This may lead them to generate more negative self-evaluations (e.g., lower self-esteem and self-recognition) and environmental expectations and appraisals (e.g., bullying, low self-control capability [53]), which increases depressive symptoms.

Depressive symptoms triggered by harsh parenting were found to aggravate suicidal ideation, providing evidence to support a previous study [81]. Based on the escape theory of suicide [45,46], adolescents who suffer from unbearable distress are likely to generate suicidal ideation to escape from painful experiences. Depressive symptoms bring adolescents dejection, low self-worth, and various painful experiences. When increased depressive symptoms exceed the scope of adolescents’ ability to withstand them, they may consider ending these experiences through suicide and hence experience enhanced suicidal ideation [45].

In addition to the above two direct relationships, it is important to pay attention to the mediation effect of depressive symptoms. Previous studies have focused mainly on the direct impacts of harsh parenting on depressive symptoms or suicidal ideation, while little is known about the mechanisms through which harsh parenting influences suicidal ideation. We found that depressive symptoms play a vital role in linking the relationship between harsh parenting and suicidal ideation, enriching the understanding of the psychological process by which harsh parenting induces suicidal ideation.

#### 4.1.2. The Moderating Effects of Mindfulness

By investigating the moderating role of mindfulness, our study revealed under what conditions harsh parenting has a strong or weak impact on suicidal ideation among adolescents. The results showed mindfulness attenuated the positive direct and indirect effects of harsh parenting on suicidal ideation. However, the moderating effect of mindfulness on the relationship between harsh parenting and depressive symptoms was not significant. One possible explanation for this finding is that adolescent mindfulness tends to have a stronger direct effect on depressive symptoms (*β =* −0.43, *p* < 0.001; see Model 1-2 in Table 3) rather than moderating the association between harsh parenting and depressive symptoms. Previous studies have also indicated that mindfulness is closely related to reduced negative emotions through effective affective regulation [31,50]. Future research is needed to further confirm the role of mindfulness in the association between harsh parenting and depressive symptoms.

This study not only provides a holistic and deeper understanding of the detrimental effects of harsh parenting but also helps identify protective factors to prevent suicidal ideation. While previous studies on mindfulness and adolescent mental health problems were mainly conducted in Western countries, this is one of the first studies to support the moderating effect of mindfulness on the relationship between harsh parenting and Chinese adolescents’ suicidal ideation, providing new guidance and ideas to alleviate suicidal ideation among Chinese adolescents.

According to the COR theory [25], adolescent mindfulness may play a crucial role in preventing the resource loss spiral caused by harsh parenting and detecting new resources. As suggested by Glomb et al. [82], mindfulness has two core functions: decoupling of the self from old experiences, and decreasing automaticity. Firstly, Chinese adolescents are largely influenced by Confucianism, and they tend to be submissive to their parents. Mindfulness can help them to act according to experience, view the culture of submissiveness in an objective manner, and better regulate emotions when experiencing harsh parenting [83]. As such, adolescents with high mindfulness are likely to separate ego from inhabitant experiences, rather than infer self-relevance, and recognize that the negative thoughts brought about by harsh parenting are replaceable instead of the only accurate representation of the facts, thereby weakening the negative effects of harsh parenting. Second, mindfulness can reduce the automaticity of emotions, defuse the tendency to respond quickly and reactively to stimuli, and support the avoidance of narrow thought [84]. This may enable adolescents to respond appropriately to harsh parenting, decrease parochialization and rumination, and reduce the effect of harsh parenting on suicidal ideation.

This study expands the current understanding of the moderating role of mindfulness in the direct and indirect associations between harsh parenting and suicidal ideation from the perspective of the COR theory [26,33]. We contribute to the mindfulness literature by showing that mindfulness, as a personal resource, buffers the influence of hash parenting on suicidal ideation via depressive symptoms. More importantly, our results suggest that, as the objects of parenting, adolescents are not just passive recipients of harsh parenting. Adolescents can proactively reduce the negative effects of harsh parenting by applying their personal resources and adopting effective strategies to cope with this family stressor.

### 4.2. Practical Implications

Suicidal ideation is a key issue in adolescent interventions, but in practice, social workers and parents have largely ignored the prevalence of harsh parenting in China and the adverse effect of harsh parenting on suicidal ideation. By investigating the underlying mechanisms and the boundary conditions of harsh parenting and suicidal ideation in the Chinese cultural context, this study offers several meaningful practical implications for educators, parents, and adolescents.

First, by confirming the detrimental effects of harsh parenting on adolescents’ depressive symptoms and suicidal ideation, we provide evidence that educators should not only pay attention to adolescents’ mental health but also focus more on parenting styles and help parents recognize the influence of their style of parenting on the parent–child relationship [20,85]. Several strategies could be used to prevent harsh parenting by Chinese parents. For example, parents may learn to handle family relationships differently, reducing unnecessary conflict and avoiding the spillover effects of family conflicts into harsh parenting. Parents could also be educated on the unique Chinese culture and intergenerational effects and learn to attach greater importance to parent–child relationships. Additionally, to improve parent–child relationships and enhance the understanding and trust between parents and children, parents need to improve their educational perspective and effectively manage their emotions [33].

Second, mindfulness is found to weaken the negative impact of harsh parenting on suicidal ideation in adolescents, as well as the indirect effect of harsh parenting on suicidal ideation via depressive symptoms. Thus, this study suggests that schools and parents should be aware of the benefits of mindfulness. Schools can provide training programs, courses, and activities to increase mindfulness in adolescents. Empirically, Dimidjian et al. revealed that Mindfulness-Based Cognitive Therapy can effectively enhance levels of mindfulness [86]. Parents can offer instruction in mindfulness for their children when children experience a sense of insecurity or a weaker attachment relationship, and help children realize that both insecurity and attachment relationships are only feelings and thoughts, and not necessarily accurate representations of reality. Finally, to protect adolescents themselves, they should be trained to understand the potential negative impact of harsh parenting and the positive impact of mindfulness on their mental health. To avoid the harmful effects of harsh parenting, adolescents can engage in mindfulness and foster the ability to focus on the present and regulate emotions.

### 4.3. Limitations and Future Research Directions

Despite the contributions of this study, there are still some limitations. Firstly, although we demonstrated the mediating role of depressive symptoms in the relationship between harsh parenting and suicidal ideation, other potential mediating mechanisms may exist. For example, adolescents who experience harsh parenting are more likely to form a low sense of self-worth and be bullied in school due to that low sense of self-worth [53]. Therefore, harsh parenting may influence adolescents’ suicidal ideation through increased exposure to bullying. To deepen our understanding of the mechanisms of harsh parenting, future studies are encouraged to explore the existence of other potential mediators that may link harsh parenting to adolescent suicidal ideation.

Secondly, this study investigated the buffering effects of mindfulness on the consequences of harsh parenting and provided important implications for coping with harsh parenting and reducing adolescent mental health problems, such as depressive symptoms and suicidal ideation. However, environmental factors, such as social support and teacher–student relationships [87,88], may also moderate the effects of harsh parenting on adolescent mental health. Therefore, future studies could explore other potentially protective factors.

Thirdly, this study focused mainly on the impacts of parenting styles on adolescents’ depressive symptoms and suicidal ideation. However, according to ecological systems theory [89,90], adolescents’ depressive symptoms and suicidal ideation may act as risk factors to induce parents’ harsh parenting and aggravate the effects of harsh parenting, causing a vicious circle and exacerbating adolescents’ mental health problems [1]. Therefore, future studies are encouraged to explore the potential interaction process relationships between harsh parenting and adolescent mental health problems.

Fourthly, this study focused on the effects of harsh parenting on adolescents’ suicidal ideation in the Chinese cultural context. It is valuable to replicate our findings in different cultural contexts. We encourage future studies to compare the impact of harsh parenting on adolescent mental health and investigate the protective effect of mindfulness in various cultural contexts. It would be an interesting topic for exploring the potential implications of parenting styles from a cross-cultural perspective.

Fifthly, our study found that mindfulness can alleviate the relationship between harsh parenting and adolescent suicidal ideation. However, we did not directly investigate whether mindfulness decreases the incidence of suicide, which could be an important outcome because the adolescent suicide rate remains an important concern for public health [12,91]. Future studies should investigate whether and how mindfulness can influence the incidence of suicide among adolescents to further extend the existing literature.

## Figures and Tables

**Figure 1 ijerph-19-09731-f001:**
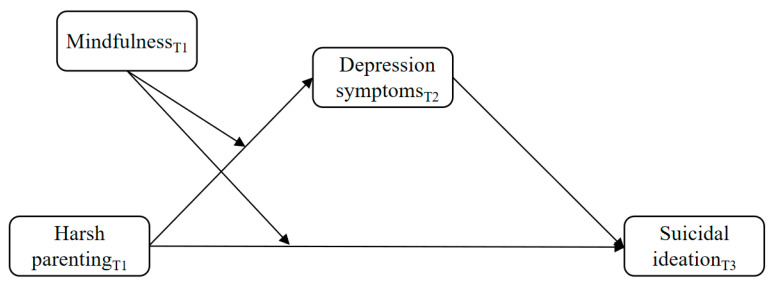
Hypothesis model of the effect of harsh parenting on suicidal ideation through depression symptoms and the moderating effect of mindfulness.

**Figure 2 ijerph-19-09731-f002:**
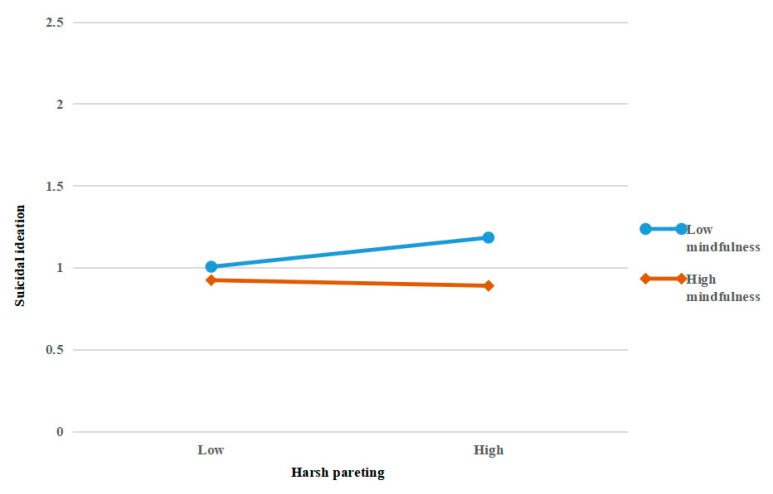
Interaction of mindfulness and harsh parenting on suicidal ideation.

**Table 1 ijerph-19-09731-t001:** Means, standard deviations, and correlations.

Variables	*Mean*	*SD*	1	2	3	4
1. Harsh parenting_T1_	1.58	0.65	-			
2. Mindfulness_T1_	3.87	1.33	−0.36 **	-		
3. Depressive symptoms_T2_	1.64	0.61	0.29 **	−0.45 **	-	
4. Suicidal ideation_T3_	1.27	0.46	0.22 **	−0.31 **	0.34 **	-

Note: ** *p* < 0.01.

**Table 2 ijerph-19-09731-t002:** Results of hierarchical regression analysis.

	Depressive Symptoms		Suicidal Ideation		
	Model 1-1	Model 1-2	Model 2-1	Model 2-2	Model 2-3
**Predictor**					
Harsh parenting (A)	0.29 ***	0.14 *	0.22 ***	0.13 *	0.02
**Mediator**					
Depressive symptoms				0.30 ***	0.24 ***
**Moderator**					
Mindfulness (B)		−0.43 ***			−0.17 **
A × B		0.01			−0.13 *
*R* ^2^	0.08	0.24	0.05	0.13	0.15
Δ*R*^2^		0.16 ***		0.08 ***	0.02 **
*F*	34.40 ***	39.27 ***	18.36 ***	27.99 ***	17.70 ***

Note. * *p* < 0.05, ** *p* < 0.01, *** *p* < 0.01. Standardized regression coefficients are reported.

**Table 3 ijerph-19-09731-t003:** Conditional indirect effects of harsh parenting on suicidal ideation through depressive symptoms.

Moderator	Level	Mediator	Outcome	Indirect Effect	Boot SE	95% CI
Mindfulness	Low	Depressive symptoms	Suicidal ideation	0.04	0.02	[0.001, 0.066]
	Medium			0.03	0.02	[0.005, 0.055]
	High			0.03	0.02	[−0.001, 0.065]

Note. CI = confidence interval.

## Data Availability

Data that support the findings of this study are available from the corresponding author upon reasonable request.

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
