# Peer review of "The Role of Mindfulness in Mitigating the Detrimental Effects of Harsh Parenting among Chinese Adolescents: Testing a Moderated Mediation Model in a Three-Wave Study"

_ijerph, 2022, doi:10.3390/ijerph19159731_

Round 1
Reviewer 1 Report
The manuscript is very well written and presented. Enough background information is given for the study and appropriate conclusion has been drawn that is well supported by the data. Authors should also refer manuscripts with PMID-28264496 and PMID-33475401 in the introduction/discussion. Overall the manuscript is well supported by the data and is significant for the field.
Author Response
Comments from Reviewers 1
Comments to the author: The manuscript is very well written and presented. Enough background information is given for the study and appropriate conclusion has been drawn that is well supported by the data. Authors should also refer manuscripts with PMID-28264496 and PMID-33475401 in the introduction/discussion. Overall the manuscript is well supported by the data and is significant for the field.
Author’s response: First of all, we are grateful for the time and effort that you have invested in helping us refine this manuscript. And thank you for providing very useful literature for us to further improve our manuscript. We have carefully read the recommended references and learned much from these references. The first paper offered strong support that mindfulness can mitigate the negative effects of stress and trauma related to adverse childhood exposures and have the potential to reduce poor health outcomes in adulthood. The second paper examined the psychometric properties of a Chinese version of Multidimensional Assessment of Parenting Scale, which facilitates future studies examining the effect of positive and negative parenting on children’s psychopathological adjustment in Chinese-speaking societies. We have added the recommended references into the introduction section of the revised manuscript. Please see the introduction section on page 3 and page 2 for full details.
Below, we have provided the revised portions of the manuscript for your reference.
(p. 3, line 134-136) Research has suggested that mindfulness may reduce the negative effects of stress and trauma associated with adverse childhood exposures, improve short- and long-term outcomes, and potentially alleviate adverse health outcomes in adulthood [49].
(p. 2, line 56-57) An example is that Ahemaitijiang et al. [22] validated both positive and negative dimensions of parenting to measure the parenting styles in the Chinese cultural context.
Ortiz, R., & Sibinga, E. M. (2017). The role of mindfulness in reducing the adverse effects of childhood stress and trauma. Children, 4(3), 16. https://doi.org/10.3390/children4030016
Ahemaitijiang, N., Han, Z. R., Dale, C., DiMarzio, K., & Parent, J. (2021). Psychometric properties of the Chinese version of the Multidimensional Assessment of Parenting Scale. Psychological Assessment, 33(3), e1. https://doi.org/10.1037/pas0000981

Reviewer 2 Report
Thank you for your submission which looks at mindfulness as a strategy to mitigate the effect of harsh parenting in Chinese adolescents, particularly suicidal ideation. It was demonstrated that harsh parenting is associated with increased suicidal ideation. Mindfulness was not statistically significant in reducing depressive symptoms but did significantly reduce suicidal ideation being both directly and indirectly related through depressive symptoms.
The paper is well written and clear, with detailed methodology and results. Of the 4 hypotheses proposed, 3 were supported and one was not. The methods ought to be reproducible and it would be interesting to see a similar study in different cultures.
Whether mindfulness decreases the incidence of suicide is not mentioned in this study but would be a useful outcome measure that would further add to the literature.
The beneficial effects of mindfulness are well known and this study demonstrated a further specific role for mindfulness.
Author Response
Comments from Reviewers 2
- Thank you for your submission which looks at mindfulness as a strategy to mitigate the effect of harsh parenting in Chinese adolescents, particularly suicidal ideation. It was demonstrated that harsh parenting is associated with increased suicidal ideation. Mindfulness was not statistically significant in reducing depressive symptoms but did significantly reduce suicidal ideation being both directly and indirectly related through depressive symptoms.
The paper is well written and clear, with detailed methodology and results. Of the 4 hypotheses proposed, 3 were supported and one was not. The methods ought to be reproducible and it would be interesting to see a similar study in different cultures.
Author’s response: We feel many thanks for your comments about our manuscript, which have helped us significantly improve our work. In this study, we discussed the underlying mechanism the and boundary condition for the relationship between parenting styles and adolescent mental health in a Chinese cultural context. In fact, as you stated, it would be interesting to see a similar study or examine the theoretical model in different cultures. Therefore, we encourage researchers to conduct cross-cultural studies to investigate and extend our theoretical model in various cultural contexts. It would be a fascinating topic for exploring the effect of parenting styles on adolescents’ mental health from a cross-cultural perspective. We have discussed this in the limitations and future research directions on page 10, line 439-443.
Below, we have provided the revised portions of the manuscript for your reference.
(p. 10, line 439-443) Fourth, this study focused on the effects of harsh parenting on adolescent’ suicidal ideation in the Chinese cultural context. It is valuable to replicate our findings in different cultural contexts. We encourage future studies to compare the impact of harsh parenting on adolescent mental health and investigate the protective effect of mindfulness in various cultural contexts. It would be an interesting topic for exploring the potential implications of parenting styles from a cross-cultural perspective.
- Whether mindfulness decreases the incidence of suicide is not mentioned in this study but would be a useful outcome measure that would further add to the literature.The beneficial effects of mindfulness are well known and this study demonstrated a further specific role for mindfulness.
Author’s response: Thanks for pointing out this. In this study, we focused on the mitigating effect of mindfulness on the relationship between harsh parenting and suicidal ideation, since suicidal ideation has been considered one of the major factors that may cause the suicide in adolescents. Suicidal ideation is an invasive and repetitive way of thinking, which is highly likely to lead adolescents to transform abstract ideas into specific plans (Kothgassner et al., 2021; Lewinsohn et al., 1996; Luby et al., 2019).
Nevertheless, we did not investigate whether mindfulness decreases the incidence of suicide, which can be a very useful outcome measure under the current theoretical framework. Suicide is defined as the act of intentionally causing one’s death in anticipation or with awareness of the consequences. Suicide among youth is a main cause of mortality worldwide, and the second cause in the 15–24 age range (Khan et al., 2018; Forte et al., 2021). Therefore, it is of significance to investigate whether mindfulness can alleviate the incidence of suicide among adolescents. To further add to the existing literature, future studies are encouraged to include the adolescent suicide rate as an important outcome when investigating the topic of mindfulness. We have described this in the limitations and future research directions section on page 10, line 444-448.
Below, we have provided the revised portions of the manuscript for your reference.
(p. 10, line 444-448) Fifth, our study found that mindfulness can alleviate the relationship between harsh parenting and adolescent suicidal ideation. However, we did not directly investigate whether mindfulness decreases the incidence of suicide, which could be an important outcome because the adolescent suicide rate remains an important concern for public health (Bridge et al., 2006; Luby et al., 2019). Future studies should investigate whether and how mindfulness can influence the incidence of suicide among adolescents to further extend the existing literature.
Bridge, J. A., Goldstein, T. R., & Brent, D. A. (2006). Adolescent suicide and suicidal behavior. Journal of child psychology and psychiatry, 47(3‐4), 372-394.
https://doi.org/10.1111/j.1469-7610.2006.01615.x
Forte, A., Orri, M., Turecki, G., Galera, C., Pompili, M., Boivin, M., ... & Geoffroy, M. C. (2021). Identifying environmental pathways between irritability during childhood and suicidal ideation and attempt in adolescence: findings from a 20-year population-based study. Journal of Child Psychology and Psychiatry, 62(12), 1402-1411.
https://doi.org/10.1111/jcpp.13411
Khan, S. Q., de Gonzalez, A. B., Best, A. F., Chen, Y., Haozous, E. A., Rodriquez, E. J., ... & Shiels, M. S. (2018). Infant and youth mortality trends by race/ethnicity and cause of death in the United States. JAMA pediatrics, 172(12), e183317-e183317. https://doi.org/10.1001/jamapediatrics.2018.3317
Kothgassner, O. D., Goreis, A., Robinson, K., Huscsava, M. M., Schmahl, C., & Plener, P. L. (2021). Efficacy of dialectical behavior therapy for adolescent self-harm and suicidal ideation: a systematic review and meta-analysis. Psychological Medicine, 1-11. https://doi.org/10.1016/j.jad.2021.04.073
Lewinsohn, P. M., Rohde, P., & Seeley, J. R. (1996). Adolescent suicidal ideation and attempts: prevalence, risk factors, and clinical implications. Clinical Psychology: Science and Practice, 3(1), 25-46. https://doi.org/10.1111/j.1468-2850.1996.tb00056.x
Luby, J. L., Whalen, D., Tillman, R., & Barch, D. M. (2019). Clinical and psychosocial characteristics of young children with suicidal ideation, behaviors, and nonsuicidal self-injurious behaviors. Journal of the American Academy of Child & Adolescent Psychiatry, 58(1), 117-127. https://doi.org/10.1016/j.jaac.2018.06.031

Reviewer 3 Report
Thank you for allowing me to review the manuscript titled "The role of mindfulness in mitigating the detrimental effects of 1 harsh parenting among Chinese adolescence: Testing a moder- 2 ated mediation model in a three-wave study" by Sun et al.
Comments:
Line 29-31: Grammatically wrong sentence. Improvise
Line 33-34: Change the flow of sentence
Line 36-38: Wrong wording again
Line 44: Low et al should be there and not just "Low"
Line 55: Change the word white racial families. Can be interpreted wrongly
Line 60: Wording needs to be corrected again
Sine there a lot of work that needs to be done on grammar and spelling, I am refraining on commenting further on it
Authors say middle schools and report ages 12 -16? Please explain
Hypothesis testing and statistical analysis are sound
Author Response
Comments from Reviewers 3
Comments to the Author: Thank you for allowing me to review the manuscript titled "The role of mindfulness in mitigating the detrimental effects of harsh parenting among Chinese adolescence: Testing a moderated mediation model in a three-wave study" by Sun et al.
- Line 29-31: Grammatically wrong sentence. Improvise
Author’s response: Thank you so much for your excellent comments about our manuscript. All of your comments are very insightful and helpful to improve the first manuscript. We had tried to revise point by point based on your comments. Based on your comments, we have revised this sentence as follow.
(p. 1, line 28-31): The literature indicates that suicidal behavior [2], depressive symptoms [3], and aggression are common mental health problems in adolescence [4], among which suicidal behavior has been widely discussed by scholars and practitioners because of its high case-fatality ratio [5].
- Line 33-34: Change the flow of sentence
Author’s response: Thanks for pointing out this. According to your comments, we have changed the flow of sentence as follow.
(p. 1, line 33-35): In order to effectively prevent and control adolescent suicide, it is necessary to investigate this topic in China, which is an Eastern cultural context with a large population [8].
- Line 36-38: Wrong wording again
Author’s response: We are very sorry for our mistakes. We have revised this sentence as follow.
(p. 1, line 36-38): Suicidal ideation is an invasive and repetitive way of thinking regarding death or harming oneself, which is highly likely to lead adolescents to transform abstract ideas or thoughts into specific plans [9-11].
- Line 44: Low et al should be there and not just "Low"
Author’s response: Thanks for pointing out this. We have checked this reference and found that “Low” is the single author of this reference. We provided the resource of this reference below.
Low, Y. T. A. (2021). Family conflicts, anxiety and depressive symptoms, and suicidal ideation of Chinese adolescents in Hong Kong. Applied Research in Quality of Life, 16(6), 2457-2473. https://doi.org/10.1007/s11482-021-09925-7
- Line 55: Change the word white racial families. Can be interpreted wrongly
Author’s response: We fully agree with your opinion. Based on your comments, we have deleted the word of white racial families as follow.
(p. 1, line 53-54): However, most of the relevant research on harsh parenting has focused on the families in the United States or other Western countries.
- Line 60: Wording needs to be corrected again.
Author’s response: Thank you for your reminder. Following your comments, we have corrected this sentence as follow.
(p. 1, line 58-59): Therefore, the generalizability of the link between harsh parenting and suicidal ideation in Eastern cultural contexts needs to be further tested.
- Sine there a lot of work that needs to be done on grammar and spelling, I am refraining on commenting further on it.
Author’s response: We apologize for the poor language of our manuscript. We worked on the manuscript for a long time and checked carefully to correct mistakes regarding grammar and spelling. We also received professional editing service for improving our manuscript. We hope that the flow and language level have been substantially improved. Please see our revised manuscript.
- Authors say middle schools and report ages 12 -16? Please explain. Hypothesis testing and statistical analysis are sound.
Author’s response: In general, the age of students in middle schools in China ranges from 12 to 18. However, this may be different in other countries. Thus, we have changed the term of “middle school” to “junior high school” to more accurately describe the participants’ grades. Please see page 7 line for full details.
(p. 4, line 192-193) The present study recruited 470 adolescents from two junior high schools in central China, using a convenience sampling method.
